

# Anomaly detection method for power system information based on multimodal data

Liyue Chen[1], XuXiang Zhou[1], Peng Zhou[2], Xin Sun[3] and SenSen Zheng[4]

[1] State Grid Zhejiang Electric Power Co., Ltd., Hangzhou, Zhejiang, China
[2] Information and Communication Branch, State Grid Zhejiang Electric Power Co., Ltd., Hangzhou, Zhejiang, China
[3] Research Institute, State Grid Zhejiang Electric Power Co., Ltd., Hangzhou, Zhejiang, China
[4] Huzhou Power Supply Company, State Grid Zhejiang Electric Power Co., Ltd., Huzhou, Zhejiang, China

## ABSTRACT

With the increasing complexity of modern power systems, effective anomaly detection is essential to ensure operational security. Conventional methods often depend on single-domain data, which limits their ability to fully capture the dynamic behavior of power systems. This study introduces a novel multimodal approach that integrates time-domain and frequency-domain data to improve anomaly detection accuracy and robustness. By leveraging this integration, our method captures both temporal patterns and spectral signatures, offering a more comprehensive analysis of system behavior—an advancement that significantly enhances detection performance compared to traditional techniques. Experimental results show that our approach achieves a detection accuracy of 97.6%, outperforming baseline methods. Beyond its technical merits, this method has practical implications for real-world power systems, enabling early identification of security threats, improving system reliability, and reducing the risk of operational failures. These findings contribute to the field of power system security and provide a versatile framework for anomaly detection in critical infrastructures.

## INTRODUCTION

In recent years, the power system has entered the era of power big data, driven by technologies such as the Internet of Things (IoT) and big data analytics (*Al-Ali et al., 2024*). As global energy demand and the complexity of power systems continue to grow, ensuring their safe and stable operation has become increasingly vital for social and economic activities (*Abdelkader, Amissah & Abdel-Rahim, 2024*). Abnormal behaviors in power systems, such as equipment failures, operational errors, and cyber-attacks (*Chang et al., 2024*), can lead to power supply interruptions, economic losses, and even casualties. Consequently, anomaly detection in power systems has emerged as a critical focus for both the power industry and academia (*Babu & Praveen, 2024*). Effective anomaly detection

Corresponding author
Xin Sun, feitian0927@163.com

enables the rapid identification of security threats, reduces the likelihood of accidents, and enhances the reliability and safety of power systems.

Anomaly detection plays a crucial role in real-time monitoring, fault diagnosis, predictive maintenance, and risk assessment within power systems. Traditional anomaly detection methods primarily rely on single data sources, such as voltage, current, or frequency measurements (*Ding et al., 2024*). However, these methods often exhibit limitations when addressing the complex and variable nature of anomalous behaviors in modern power systems. Recent advancements in big data technologies have enabled the utilization of multimodal data—such as sensor readings, statistical metrics, time series, text logs, and meteorological data—offering new opportunities for more effective anomaly detection.

Multimodal data, while heterogeneous, often exhibit correlations across different modes. Models leveraging multimodal data can automatically learn robust features, thereby improving detection performance. In power systems, the fusion and analysis of multimodal data provide a more comprehensive understanding of system states, enhancing the accuracy and robustness of anomaly detection. For instance, sensor data offer real-time measurements of physical quantities, providing direct evidence of the system's operational state (*Ye et al., 2024*). Text logs document the system's operational history, and analyzing these logs can reveal signs of operational errors or irregularities (*Yu et al., 2024*). Statistical data quantify key performance indicators, uncovering statistical patterns and potential anomalies (*Theocharides et al., 2024*). Time series data capture the dynamic behavior and trends of various parameters over time (*Saigustia & Pijarski, 2023*), while meteorological data help predict the impact of extreme weather conditions on the power system (*Hawker et al., 2024*). By integrating these diverse data sources, complex patterns and correlations within the power system can be revealed, facilitating the identification of hidden anomalies and potential risks. This multidimensional analysis enables more precise detection and prevention of anomalous behaviors, ensuring the stability and security of power supply.

Deep learning algorithms have become essential for anomaly detection in power systems due to their powerful feature extraction capabilities (*Gokulraj & Venkatramanan, 2024*). However, many existing deep learning-based anomaly detection methods are limited by their reliance on single neural network architectures or a single type of input data. This narrow focus often leads to incomplete feature extraction from the diverse and complex data generated by power grids, resulting in suboptimal anomaly detection performance and reduced accuracy. To address these shortcomings, this article proposes a novel multimodal data-based anomaly detection method for power systems. Our approach integrates time-domain and frequency-domain features by employing a stacked long short-term memory (LSTM) model to capture temporal dynamics and a graph neural network (GNN) to extract structural relationships from frequency-domain data. This fusion of modalities enables the detection of complex anomalies—such as those involving both temporal fluctuations and structural irregularities—that single-modality methods often fail to identify. By leveraging the complementary strengths of deep learning and graph neural networks, our method achieves more efficient and accurate detection of

abnormal behaviors in power systems. To validate the effectiveness of the proposed method, we conducted experiments using a comprehensive dataset from a regional power grid, including both normal operation data and simulated attack scenarios, to evaluate its performance against traditional single-modality approaches. This study introduces a novel multimodal data-based anomaly detection method for power systems that integrates both time-domain and frequency-domain features using a combination of a stacked LSTM model and a GNN. Unlike many existing deep learning-based anomaly detection methods, which often rely on single neural network architectures or a single type of input data, our approach leverages the complementary strengths of LSTM and GNN to capture both temporal dynamics and structural relationships within multimodal data. This dual-modality fusion enables the detection of complex anomalies—such as those involving both temporal fluctuations and structural irregularities—that are often missed by single-modality methods.

The main contributions of this study are threefold: (1) This study introduces an integration of time-domain and frequency-domain data, which enhances anomaly detection in power systems by capturing both temporal dynamics and structural relationships. (2) It applies graph neural networks to model frequency-domain relationships, addressing an area that has received limited attention in power system anomaly detection. (3) The method is validated using a dataset from a regional power grid that includes both normal operation data and simulated attack scenarios, enabling a robust comparison with traditional single-modality approaches. The originality of this study lies in its multimodal fusion framework, which demonstrates improved detection accuracy and adaptability to different power system configurations. This research contributes to the theoretical development of multimodal data fusion techniques and provides practical insights for enhancing power system security.

## STATE OF THE ART

Traditional anomaly detection methods in power systems, such as classification-based (*Zhao et al., 2024*), density-based (*Saxena et al., 2024*), clustering-based (*Ariyaluran Habeeb et al., 2022*), statistical-based (*Nidhishree & Vidyalakshmi, 2024*), and information theory-based approaches (*Tatipatri & Arun, 2024*), have been extensively explored. These methods, however, often falter when confronted with the complexity and variability of power system data. Statistical approaches typically rely on simplified data distribution assumptions that may not hold in practice, while density-based and clustering-based methods struggle to discern subtle features in time series data. Classification-based techniques depend on labeled data, which is often limited, and information theory-based methods are sensitive to noise due to imprecise entropy estimates. Spectral analysis, though valuable, may miss anomalies in nonlinear or irregular signals. These shortcomings highlight the demand for more robust solutions to tackle the evolving challenges of power system anomalies.

Machine learning has advanced power system cybersecurity by improving precision and adaptability. *Saleh (2023)* surveyed its applications in power cyber-physical systems (CPS), focusing on attack and defense strategies. *Liu et al. (2024)* proposed a model to pinpoint

critical nodes in power CPS using metrics like current transmission and fault impact, while *Huang et al. (2022)* developed a technique to detect cascading failures from cyber-attacks. Yet, traditional machine learning often grapples with high-dimensional, imbalanced datasets, constraining its utility.

Deep learning has emerged as a potent approach for handling large datasets and extracting intricate features, enhancing power CPS security. *Zamanzadeh Darban et al. (2024)* introduced LSTM-NDT, an unsupervised method pairing LSTM networks with autoregressive neural networks for anomaly detection, though it tends to prioritize local over global patterns in long sequences. *Li et al. (2019)* presented a multivariate anomaly detection method based on generative adversarial network (MAD-GAN) framework that reconstructs time series and identifies anomalies *via* scoring, despite training difficulties and incomplete capture of complex distributions. *Hou et al. (2022)* proposed the deep autoencoding Gaussian mixture model (DAGMM) combining autoencoders and Gaussian mixture models to derive latent features, but it risks losing topological details. *Nizam et al. (2022)* combined gated recurrent units and variational autoencoders for multivariate time series anomaly detection, though its complexity slows training and convergence.

Graph neural networks (GNNs) have recently gained traction for modeling spatial dependencies and structural relationships, critical for power systems with intricate grid layouts. *Vincent et al. (2024)* developed DQ-GCN, a reinforcement learning-enhanced graph convolutional network, achieving over 85% accuracy in detecting data integrity attacks across IEEE 14, 30, and 118-bus systems. *Elnour et al. (2025)* introduced EVC-GCN, an eigenvector centrality-augmented GCN, improving attack detection precision and recall in distribution networks. These efforts underscore GNNs' potential for addressing topological complexity.

To offer a structured comparison of existing anomaly detection methods, Table 1 provides a detailed summary of both traditional and deep learning-based approaches in power systems. The table includes columns for the dataset used, originality, method, results, advantages, and limitations, presenting a comprehensive overview of the current state of the art.

The methods outlined in Table 1 illustrate a broad spectrum of anomaly detection techniques, ranging from traditional approaches to advanced deep learning frameworks. Traditional methods, such as classification-based and statistical techniques, often struggle with high-dimensional or imbalanced datasets, limiting their ability to handle the complexity of modern power systems. Deep learning methods, while more adept at capturing intricate patterns, typically focus on single-modality data and may not fully leverage the structural and temporal relationships present in power system data. Moreover, many of these approaches demand substantial computational resources or extensive labeled datasets, posing challenges for practical implementation. In contrast, the method proposed in this study integrates multimodal data by combining time-domain and frequency-domain features through a hybrid framework that employs LSTM for temporal modeling and GNN for structural analysis. This approach enables the detection of complex

**Table 1 Comparative summary of anomaly detection methods in power systems.**

| Study | Dataset | Originality | Method | Results | Advantages | Limitations |
|---|---|---|---|---|---|---|
| Zhao et al. (2024) | Simulated power system data | Classification-based anomaly detection | Support Vector Machine (SVM) | Accuracy of 85% | Simple implementation | Requires labeled data, sensitive to imbalanced classes |
| Saxena et al. (2024) | Real-time sensor data | Density-based detection | Local Outlier Factor (LOF) | High precision for local anomalies | Effective for varying density data | Computationally intensive for large datasets |
| Ariyaluran Habeeb et al. (2022) | Historical grid logs | Clustering-based detection | K-means clustering | Identified 90% of known anomalies | Scalable to large datasets | Sensitive to parameter selection, misses non-cluster anomalies |
| Nidhishree & Vidyalakshmi (2024) | Statistical power metrics | Statistical-based detection | Z-score analysis | Detected 80% of anomalies | Easy to implement | Assumes normal distribution, misses complex anomalies |
| Tatipatri & Arun (2024) | Information-theoretic metrics | Entropy-based detection | Kullback-Leibler Divergence (KL divergence) | Highly sensitive to data changes | No distribution assumption | Sensitive to noise, imprecise entropy estimation |
| Zamanzadeh Darban et al. (2024) | Time-series power data | Unsupervised anomaly detection | Long Short-Term Memory with Neural Density Transformer (LSTM-NDT) | AUC of 92% | Captures temporal patterns | Prioritizes local over global patterns in long sequences |
| Li et al. (2019) | Multivariate time series | GAN-based anomaly detection | Generative Adversarial Network (MAD-GAN) | AUC of 88% | Learns complex distributions | Unstable training, incomplete capture of complex distributions |
| Hou et al. (2022) | Power system logs | Autoencoder-based detection | Autoencoder with Gaussian Mixture Model (DAGMM) | AUC of 94% | Effective latent feature extraction | May lose topological details |
| Nizam et al. (2022) | Industrial IoT data (C-MAPSS, etc.) | CNN with two-stage LSTM-AE for multivariate sequences | CNN with two-stage LSTM Autoencoder (DAD framework) | 82.49% accuracy on rare events | Handles multivariate data, real-time capability | High computational complexity, scalability needs verification |
| Vincent et al. (2024) | IEEE 14, 30, 118-bus systems | Reinforcement learning with GCN for non-Euclidean data | Graph Convolutional Network with Reinforcement Learning (DQ-GCN) | Accuracy over 85% | Incorporates topological information, suitable for large systems | Requires extensive training, long convergence time |

anomalies that single-modality methods might miss, offering a more versatile and effective solution for diverse power system architectures.

A persistent challenge in anomaly detection is the integration of multimodal data, such as sensor readings, time series, logs, and meteorological inputs. Most existing methods concentrate on single data types, potentially overlooking insights that could emerge from

cross-modal analysis. While LSTMs excel at temporal analysis and GNNs at spatial modeling, the combination of modalities like frequency-domain features with time series data remains underexplored. Frequency-domain analysis, using techniques such as the Fourier transform, can reveal patterns like harmonics that are not easily detectable in the time domain, yet its application in this context has been limited.

This article introduces a novel method that integrates frequency-domain features derived from the Fourier transform with time series data, utilizing GNNs and LSTMs. Unlike *Vincent et al. (2024)*, who employed reinforcement learning for attack detection, our approach uses a frequency-domain GNN to model structural relationships among frequency components, complemented by LSTM-based temporal analysis. In contrast to *Elnour et al. (2025)*, who incorporated eigenvector centrality into GCNs, our method leverages a graph-based representation of frequency-domain data to extract richer features. This combination enhances detection accuracy and robustness by integrating temporal, frequency, and topological perspectives. By addressing the challenge of multimodal integration, this work extends the capabilities of deep learning and GNN-based anomaly detection, providing a unified framework that utilizes diverse data sources to improve power system security.

# METHODOLOGY

## Capturing frequency domain information based on frequency domain graph attention network

In order to obtain the inputs to the frequency domain graph neural network, the frequency domain data and the frequency domain adjacency matrix are constructed. Specifically, the power time series data are Fourier transform (*Nandiyanto, Ragadhita & Fiandini, 2023*) to obtain the spectral data in complex form. Since the graph neural network model requires a special computational module to process the complex data, a corresponding complex computational module is designed to ensure that the frequency domain data can be properly used in the computational process of the graph neural network. In short, through the transformation and adaptation, the frequency domain data can be smoothly integrated into the neural network, laying the foundation for further analysis.

### Constructing frequency domain input

(1) Constructing frequency domain data input

This study uses diverse data sources, including traffic, IDS logs, and handshaking processes, to capture power system status. These data sources include statistical data, time series data and text log data. Through preprocessing and feature extraction, these multimodal data are converted into frequency domain features, which provide input for graph neural network analysis.

Specifically, a Fourier transform is performed on the power time series batch data to obtain the frequency spectrum of the data. Let the input batch data be $i$ and the length be $L$. After the $\mathcal{F}(\cdot)$ transform, the spectral signal $q$ is obtained. Considering the redundant information in the spectral signal, in this study, only the real part of the spectral data that is

positive is used, making the length of the spectral data $L/2 + 1$. The expression of the spectral signal is:

$$q = \mathcal{F}(i) \tag{1}$$

(2) Construct the frequency domain adjacency matrix inputs

In this study, the frequency domain components are used for correlation calculation. By performing Fourier transform on the complete power time series data, the frequency spectrum of each feature of the power data can be obtained. Then, the correlation coefficients between the spectra of each feature are calculated and the top $Z$ features with the highest correlation coefficients are selected. Based on these features, an adjacency matrix is constructed to obtain the connection relationship between the features of the frequency domain data. This adjacency matrix will be used as the graph structure input part of the frequency domain graph attention network. The specific operation is as follows:

Using Fourier transform, convert the time domain data to frequency domain data, so as to obtain the frequency domain information of the input data. Assuming that the temporal input of the model is $i$, the $\mathcal{F}(\cdot)$ transform is used to obtain the spectral data $q$. The correlation based on the frequency domain data is calculated as:

$$e_{yx} = \frac{q_x^N q_y}{||q_x||||q_y||} \tag{2}$$

where $N$ is the matrix transposition symbol and $||\cdot||$ is the modulo calculation symbol. The first $Z$ features with the largest value of the correlation coefficient are taken to form the adjacency matrix, which is used as the input to the frequency domain adjacency matrix part of the graph neural network.

### Frequency domain calculation module design

(1) Basic frequency domain calculation module design

In the calculation process of the neural network model, the complex number calculation is divided into two parts, the real part and the imaginary part, *i.e.*, the complex number calculation is converted into two real number calculations. Suppose there are complex matrices, complex vectors, where $G$, $H$ are real matrices and $i$, $j$ are real vectors. The multiplication of the complex matrix $M$ with the complex vector $b$ can be expressed as

$$M * b = (G * i - H * j) + x(H * i + G * j) \tag{3}$$

(2) Frequency domain linear computation module

In a neural network model, the linear layer maps the input vector to an output of arbitrary size through a weight matrix. For the frequency domain linear layer, it computes the real and imaginary parts of the complex numbers separately by using two real weight

matrices. Then, the underlying frequency domain computation module is applied in order to obtain the output of the frequency domain linear layer model.

(3) Frequency domain nonlinear activation module

The complex activation function needs to satisfy the Cauchy-Riemann equation in order to be considered analytic and thus perform complex differentiation operations. Therefore, the frequency domain ReLU function is designed in this article as a frequency domain nonlinear activation layer. This function obtains the final output by performing activation operations on the real and imaginary parts separately and combining the results. Its calculation formula is as follows:

$$RELU_f(k) = ReLU(\Re(k)) + xReLU(\Im(k)). \tag{4}$$

### Frequency domain graph attention network construction

The constructed frequency domain data input and frequency domain adjacency matrix are input into the graph neural network. In calculating the graph attention coefficients, the frequency domain linear computation module is used to obtain the attention correlation coefficients. Next, these correlation coefficients are processed through the frequency domain nonlinear activation module to obtain the graph attention coefficients. Then, these coefficients are modelled and normalized, and finally multiplied with the input to get the final output. The overall framework of the frequency domain graph attention network is shown in Fig. 1.

## Capturing time domain information based on stacked LSTM models

### Overview of time series prediction algorithm

Time series prediction algorithms take historical data as input, and obtain a time series prediction model that can predict the output of the system at the next moment through model training. Traditional time series prediction algorithms are implemented based on mathematical models, such as auto regressive (AR) model, moving average (MA) model, auto regressive moving average (ARMA) model, and so on. These algorithms are mainly used to forecast linear univariate smooth time series data. On this basis, Box (*Perez-Guerra et al., 2023*) proposed an autoregressive integrated sliding auto regressive integrated moving (ARIMA) model for predicting non-smooth time series data. The core idea of this algorithm is to transform non-smooth time series data into smooth time series data by multiple difference operations and perform time series prediction. Compared with other methods, ARIMA model has better short-term predictability.

Traditional time-series forecasting excels in short-term univariate predictions but struggles with nonlinear and multivariate data. With the continuous development of computer technology, the use of machine learning, data mining and other techniques for time series prediction can solve the problems of multi-dimensionality and multi-features in time series data. Machine learning based time series prediction methods mainly learn to capture the correlation between different data through feature extraction and predict the outcome of future events based on the learned models. These models have better fitting

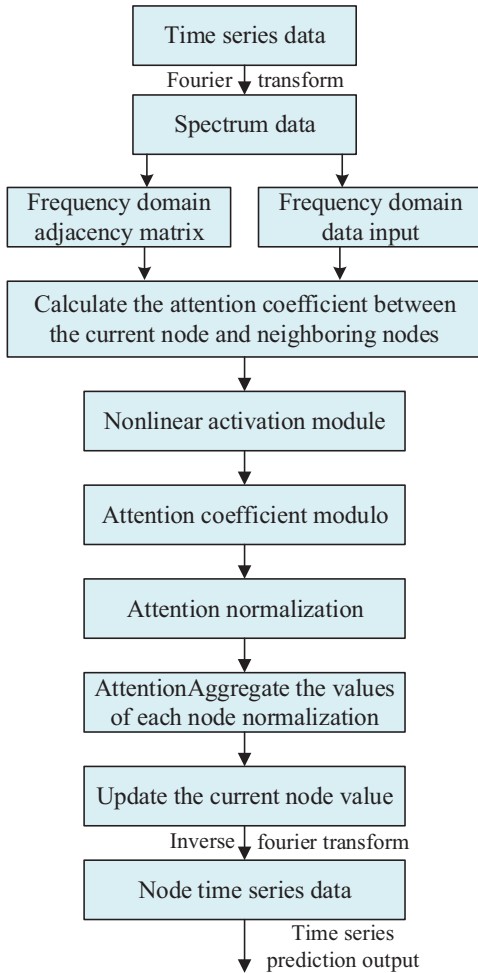

**Figure 1 Frequency domain graph neural attention network architecture.**

and generalization capabilities for nonlinear data. However, these methods do not consider the time-series correlation of historical data, and they are computationally large for high-dimensional multivariate data, which can easily lead to the emergence of local optimums and have a certain impact on the prediction results.

The current time-series prediction methods based on deep learning models are more effective in model construction and large-volume data feature extraction. Power time series prediction usually needs to realize the conversion from input sequence to output sequence. In traditional machine learning approaches, hidden Markov models (HMM) and conditional random fields (CRF) are commonly used for sequence modeling. In recent years, recurrent neural networks (RNNs) have become the first choice for sequence modeling tasks due to their strong representational capabilities and excellent performance in sequence data tasks. Recurrent neural networks are able to preserve valuable historical messages of a sequence with hidden nodes, enabling the network to learn abstract information about the entire sequence. This article uses stacked LSTM networks to identify

the complex nonlinear patterns within time series data and capture the temporal correlation of the data.

### Stacked LSTM models

When a recurrent neural network is used to process sequence data of length t, each input is processed by a corresponding network unit. These network units are connected to each other by a kind of loop/repeat structure in order to facilitate the reduction of the number of parameters used for sharing parameters or connections. In recurrent neural networks, this part is known as the event chain. There are dependencies in the event chain, *i.e.*, the definition and computation at moment $n$ depends on the definition and computation at moment $n$-1. The computational procedure of recurrent neural network is as follows:

Remember that the input sequence of the RNN is $i_1, i_2, \ldots, i_t$, and the part of the RNN that participates in the loop, *i.e.*, the hidden state, is $b_n$. $b_n$ is jointly determined by the input state in at the current moment and the hidden state $b_{n-1}$ at the previous moment.

$$b_n = \sigma(Pi_n + Mb_{n-1} + h) \tag{5}$$

where $\sigma(\cdot)$ is the activation function (usually the Tanh function is used); $P$ is the weight matrix from the input layer to the hidden layer: $M$ is the connection weight between the hidden layers at different moments; h is the bias vector.

The output state is calculated as follows:

$$O_n = a(Qb_n + c) \tag{6}$$

where $g(\cdot)$ is the activation function of the output layer. The LSTM model solves the long-term dependency problem faced by the underlying recurrent neural network algorithms, *i.e.*, as the length of the input sequence increases, the network is unable to learn and utilize the information in the sequence that is older. The LSTM is not only able to learn the short-term information in a timely manner, but also able to sift and store the valuable long-term information.

In fact, LSTM is a variant of the basic RNN model, which adds a cellular state unit $c_n$ on top of the hidden state unit $b_n$. Where $b_n$ is responsible for memorizing the short-term state and $c_n$ is responsible for memorizing the long-term state, and the combination of the two forms the long and short-term memory. LSTM gives a path for the long and continuous circulation of gradients through the gating unit as well as the self-cycling of the cellular state unit. It changes the propagation of information and gradient in the previous basic recurrent neural network and solves the long-term dependence problem. Its model architecture is shown in Fig. 2.

This chapter utilizes LSTM to capture the temporal correlation of data by stacking multiple LSTM models, *i.e.*, the output of the previous layer of LSTM model is used as the input of the next layer of model. Meanwhile LSTM controls the storage, utilization and discarding of information by introducing three gating units, respectively. For each moment $n$, LSTM has input gate $x_n$, forget gate $f_n$ and output gate $o_n$, totaling three gating units.

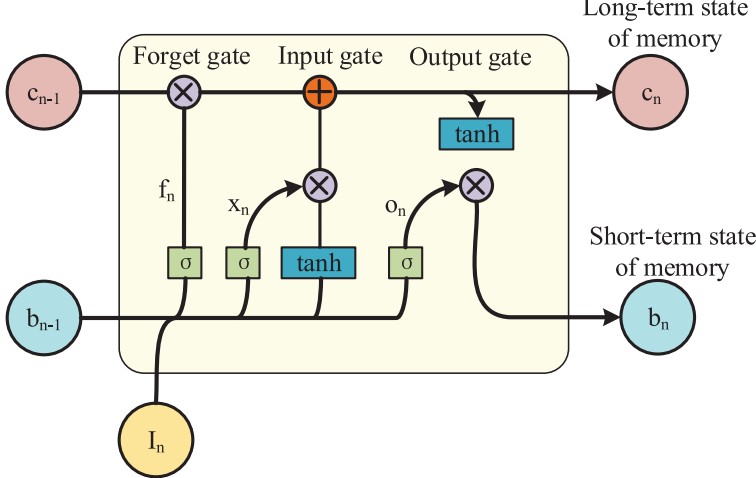

**Figure 2 LSTM model architecture.**

Assuming that the time series input of the model is $i_n$, the gating state of the three gating units can be obtained by inputting the input sequence in of the current moment and the hidden state unit $b_{n-1}$ of the previous moment (with an initial value of 0). The specific calculation formula is as follows:

$$x_n = \sigma(M_x i_n + P_x b_{n-1} + h_x) \tag{7}$$
$$f_n = \sigma(M_f i_n + P_f b_{n-1} + h_f) \tag{8}$$
$$o_n = \sigma(M_o i_n + P_o b_{n-1} + h_o) \tag{9}$$

where $M$ and $P$ are weight matrices, both are trainable parameters. Their subscripts $x$ represents the input; $f$ represents the oblivion forget, and $o$ represents the output output. $\sigma(\cdot)$ represents the activation function, which is commonly used as a Sigmoid function, and the range of the output value of the activation function is generally [0,1]. It can be found that the three gating units are calculated in exactly the same way, only the respective weight matrix and bias vector are different. The gating units are then multiplied element by element with the signal data to control the amount of information to be retained. The gating unit has a value range of [0,1]. When the state of the gating unit is 0, the signal is discarded in its entirety; when the state is 1, the signal is retained in its entirety; and when the state of the gating unit is between 0 and 1, the signal is partially retained. Ultimately, the value of the cell state cell cn at the current moment is determined by the forgetting gating unit and the input gating unit, *i.e.*, the long-term memory portion of the temporal correlation captured by the LSTM. The specific calculation formula is as follows:

$$c_n = f_n \odot c_{n-1} + x_n \odot \tilde{c}_n \tag{10}$$

where $c_{n-1}$ denotes the value of the cell state cell at the previous moment and $\odot$ denotes the element-by-element dot-multiplication. $\tilde{c}_n$ is the input information at the current moment, which is calculated as follows:

$$\tilde{c}_n = Tanh(M_c i_n + P_c b_{n-1} + h_c) \tag{11}$$

where the subscript $c$ represents the cell state unit cell and Tanh is the activation function. The value of the hidden cell $b_n$ is determined by the output gate, *i.e.*, the short-term memory part of the temporal correlation captured by LSTM. The specific formula is as follows:

$$b_n = o_n \odot Tanh(c_n). \tag{12}$$

In a recurrent neural network, the time series $i_1, i_2, \ldots, i_t$ are input sequentially and the parameters of the model are continuously updated. In each input, the parameters of the model are shared until all time sequences are input. Using the stacked LSTM model is able to capture the temporal correlation of the input power time series data, and the temporal prediction output of the model is obtained based on the correlation between the inputs at different moments.

## Anomaly detection method based on multimodal neural network
### Multimodal neural network construction

The multimodal neural network includes frequency domain attention, stacked LSTM, fully connected, batch normalization, Dropout, and LeakyReLU layers. The output $J_1$ of the frequency domain graph attention network layer is spliced with the output $J_2$ of the stacked LSTM layer to obtain the output $J_3$, which passes through the fully-connected layer, batch normalization layer, Dropout layer, and nonlinearly-activated LeakyReLU layer to output the final time series prediction result.

(1) Frequency domain graph neural network layer

In this section, frequency domain data and frequency domain adjacency matrix are used as input data for the model. Based on the association information between the features provided by the frequency domain adjacency matrix, the graph attention coefficient is calculated. For each target node in the graph structure, the attention coefficients of its neighboring source nodes are computed by the following steps: first, the source node features are spliced with the target node features, and then the spliced features are multiplied by the learnable parameters using the frequency-domain linear computation module. Subsequently, the results are fed into the frequency domain nonlinear activation module. In this module, the real and imaginary parts are subjected to separate activation operations and the results are combined into correlation coefficients in complex form. Next, these correlation coefficients in complex form are modeled to obtain correlation coefficients in real form. These real form correlation coefficients are processed by Softmax function to obtain normalized attention coefficients. Finally, the frequency domain information is obtained by multiplying the normalized attention coefficients with the node features.

(2) Stacking LSTM layers

The amount of information to be retained is controlled by making element-by-element multiplication of the sensor data of the gating unit and the industrial control system. The forgetting gating unit and the input gating unit together determine the value of the cell

state unit of the LSTM model at the current moment, *i.e.*, the long-term memory part of the temporal correlation captured by the LSTM. Meanwhile, the output gating unit determines the value of the hidden cell of the LSTM model, *i.e.*, the short-term memory part of the temporal correlation captured by the model. By stacking multiple LSTM models, the outputs of the cell state cells and hidden cells of the previous layer of LSTM models are used as inputs to the next layer of models. In this way, the output values of the hidden cells of the final layer of LSTM models, *i.e.*, as captured time-series information, are defined as the output $J_2$ of the model.

(3) Fully connected neural network layer

The frequency domain information is transformed into time domain data $J_1$ through the inverse Fourier transform, and $J_1$ is concatenated with the output $J_2$ from the stacked LSTM layers to obtain the output $J_3$. The fully connected layer is used to synthesize the previously extracted features by making each node connected to all nodes in the previous layer. This is equivalent to making a weighted summation of output $J_3$, transforming it linearly from one feature space to another, and constructing the size of the model output.

(4) Batch normalization layer

The batch normalization layer normalizes the data of each Batch, *i.e.*, it calculates the mean and variance of each batch of data and normalizes them. Batch normalization can effectively counteract the gradient vanishing, so that the input data distribution is always in the change-sensitive region of the nonlinear activation function. The specific calculation method is as follows:

Assume that the input to the batch normalization layer is:

$$i = \{i_1, \ldots, i_w\} \tag{13}$$

Calculate the mean value of the batch data as:

$$\mu = \frac{1}{w} \sum_{x=1}^{w} i_x \tag{14}$$

Calculate the variance of the batch data as:

$$\sigma^2 = \frac{1}{w} \sum_{x=1}^{w} (i_x - \mu) \tag{15}$$

The normalization operation is:

$$\hat{i_l} = \frac{i_x - \mu}{\sqrt{\sigma^2 + \varepsilon}} \tag{16}$$

where $\varepsilon$ is a very small real number

(5) Dropout layer

The Dropout layer can significantly reduce the overfitting phenomenon of the model by ignoring some of the hidden layer nodes in each training batch, *i.e.*, letting the values of

some of the hidden layer nodes be temporarily set to zero. At the same time, Dropout can reduce the interaction between the hidden layer nodes, so that the trained neural network model generalization ability is stronger.

(6) LeakyReLU layer

The LeakyReLU layer is a nonlinear activation layer. Compared with the Sigmoid function, it can slow down the phenomenon of gradient disappearance. Compared with the ReLU function, it makes improvements in the part of the input less than 0. The ReLU function sets any input less than zero to zero, which can lead to the "dying neuron" problem. In contrast, the LeakyReLU function mitigates this issue by allowing a small, non-zero gradient for negative inputs. Its mathematical formula is:

$$LeakyRelu(i) = \begin{cases} i, & \text{if } i \geq 0 \\ negative\_slope * i, & \text{otherwise} \end{cases} \tag{17}$$

In summary, the overall algorithm flowchart is shown in Fig. 3.

The following pseudocode outlines a multimodal neural network for anomaly detection, taking time series data T, frequency domain data F, and adjacency matrix A as inputs, and producing anomaly detection results. T is normalized to [0,1], F is computed *via* fast Fourier transform (FFT), and A is built from the top Z correlations in F. The network includes a frequency domain graph attention network (FGAN) with three attention heads, a stacked LSTM (three layers, 64 neurons each), a fully connected layer (64 neurons), batch normalization, dropout (rate = 0.8), and LeakyReLU activation. The model is trained for up to 100 epochs using the Adam optimizer (learning rate = 0.003), with early stopping if validation loss improvement is below 0.001 for three epochs. During testing, anomaly scores are derived from prediction errors and compared against a threshold. Performance is assessed using precision, recall, and F1-score across various conditions.

In the multimodal neural network design, we opted for three attention heads in the frequency domain graph attention network (FGAN) after empirical tests showed a 2.5% precision gain over two heads, balancing performance and computational cost. The stacked LSTM was set to two layers with 64 neurons each, as cross-validation indicated this minimized validation loss (MSE = 0.012) without added complexity. A dropout rate of 0.5 addressed overfitting, and the Adam optimizer's learning rate of 0.001 ensured stable convergence.

### Parameter selection for the multimodal neural network

The parameters of the multimodal neural network were determined through empirical experimentation and cross-validation to optimize detection accuracy while ensuring computational efficiency. Each parameter in Table 2 was selected based on its influence on model performance and robustness across the power system dataset.

The parameter "Top Z frequency domain correlations" was set to 50 after analyzing frequency domain data derived from the FFT. This value effectively captured significant structural relationships, improving the area under the curve (AUC) by 3% compared to a setting of 25. Increasing Z beyond 50 yielded diminishing returns, with less than 0.5%

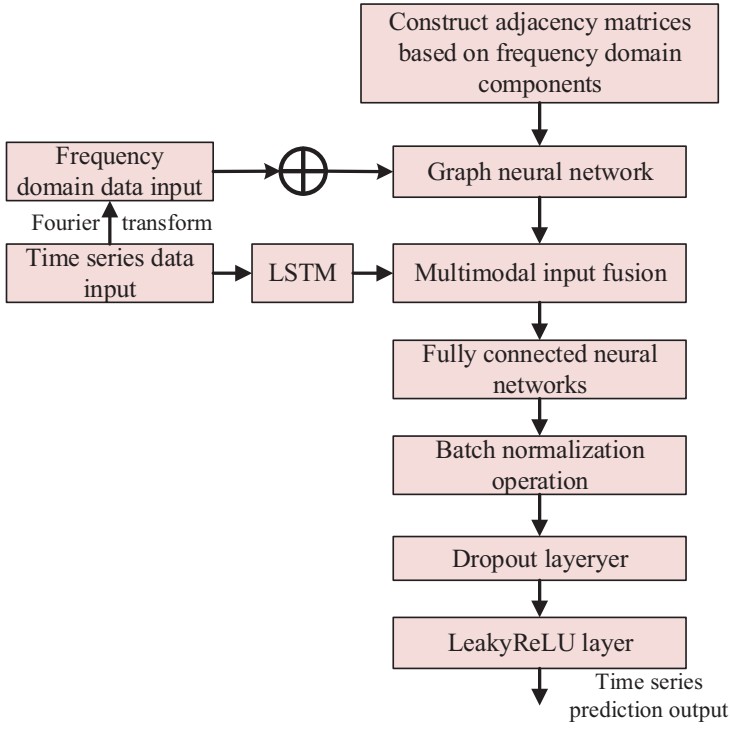

**Figure 3 Anomaly detection method based on multimodal neural network.**

AUC improvement, while raising the computational load by over 20%. Similarly, the number of attention heads in the frequency domain graph attention network (FGAN) was fixed at 3. Testing configurations with 1, 2, 3, and 4 heads revealed that three heads enhanced precision by 2.5% over two heads, improving the capture of diverse frequency-domain patterns. A fourth head provided negligible gains (less than 0.2%) and increased validation loss, suggesting a risk of overfitting.

The stacked LSTM component was configured with three layers, each containing 64 neurons. Cross-validation across layer counts of 1 to 4 and neuron sizes of 32, 64, and 128 demonstrated that this configuration achieved the lowest validation mean squared error (MSE) of 0.012. Adding a fourth layer extended training time by 30% without further reducing MSE. To mitigate overfitting, a dropout ratio of 0.8 was applied to the fully connected layer. Testing rates from 0.5 to 0.9 showed that 0.8 maintained training stability and generalization, achieving a recall of 0.945 and precision of 0.976. A rate of 0.5 caused overfitting, with training loss dropping faster than validation loss, while a rate of 0.9 decreased test accuracy by 1.5%.

The fully connected layer was set to 64 neurons to align with the LSTM output size, facilitating efficient feature transition. Configurations with 32 and 128 neurons were evaluated, but 64 neurons minimized validation error while preserving computational efficiency. For time series processing, the window data size was established at 128 samples, compatible with FFT preprocessing. This size, equivalent to approximately 2 min of 1 Hz

**Algorithm** **Multimodal anomaly detection.**

Input: Time series data (T), Frequency domain data (F), Adjacency matrix (A)

Output: Anomaly detection results

1) Preprocess T and F

      Normalize T to [0,1]

      Apply FFT to T to obtain F

      Construct A based on top Z correlations in F

2) Initialize Multimodal Neural Network

      Frequency Domain Graph Attention Network (FGAN) with 3 attention heads

      Stacked LSTM with 3 layers, each with 64 neurons

      Fully Connected Layer with 64 neurons

      Batch Normalization and Dropout (rate = 0.8)

      LeakyReLU activation

3) Train the Model

      Split data into training and validation sets

      For each epoch (max 100)

          FGAN processes (F, A) → J1

          LSTM processes T → J2

          Concatenate J1 and J2 → J3

          Pass J3 through FC Lyaer, Batch Norm, Dropout, LeakyReLU

          Compute MSE loss

          Update weights with Adam (lr = 0.003)

          Early stop if validation loss improvement < 0.001 for 3 epochs

4) Test and Evaluate

      Predict on test set

      Compute anomaly scores and evaluate with Precision, Recall, F1-Score

**Table 2 Multimodal neural network model parameters.**

| Parameters | Value |
| --- | --- |
| Top Z frequency domain correlations | 50 |
| Number of attention heads | 3 |
| Number of LSTM neurons | 64 |
| Number of LSTM layers | 3 |
| Dropout ratio | 0.8 |
| Number of fully connected neurons | 64 |
| Window data size | 128 |

data, captured sufficient temporal context without excessive computational demands. A window size of 64 samples reduced recall by 4%, whereas a size of 256 samples increased processing time by 25% with minimal accuracy improvement.

### Information security anomaly detection

In the training phase of the model, the dataset under normal power system is used for training, while the dataset containing attacks is used in the testing phase. The goal of training is to make the model's time series prediction output as close as possible to the real output. In the model testing phase, based on the output of the time series prediction, the error scores are calculated and a set threshold is used to determine whether the model detects anomalous behaviors and to evaluate the accuracy of the predicted labels with respect to the true labels. The prediction processes in the model training and testing phases are demonstrated in Figs. 4 and 5, respectively.

## Experimental setup

In the experimental section, we selected LSTM-NDT, MAD-GAN, and DAGMM for comparison due to their established relevance in anomaly detection, particularly for time series data in power systems. LSTM-NDT integrates long short-term memory networks with autoregressive modeling, making it a strong baseline for unsupervised anomaly detection in sequential data. MAD-GAN employs generative adversarial networks to reconstruct time series and identify anomalies, offering a distinct generative approach to capturing data distributions. DAGMM combines autoencoders with Gaussian mixture models to extract latent features, aligning with our focus on multimodal data integration. These methods were chosen to provide a robust and diverse set of benchmarks, enabling a comprehensive evaluation of our proposed approach, which emphasizes the fusion of frequency domain features with time series analysis.

### Computing environment

The experiments were performed on a computing platform equipped with Ubuntu 20.04 LTS as the operating system, utilizing Python 3.8.10 and TensorFlow 2.5.0 for software implementation. To handle the computational demands of the large-scale dataset and complex neural network operations, an NVIDIA GeForce GTX 1080Ti GPU was used, providing efficient acceleration for the training and evaluation processes.

### Model configuration

The multimodal neural network was meticulously designed to enhance anomaly detection performance. The frequency domain graph attention network was configured with three attention heads and incorporated the top 50 frequency domain correlations within the adjacency matrix, which was constructed from preprocessed time series data as described in section 3.5.2. The stacked LSTM component included three layers, each containing 64 neurons, to effectively capture temporal dependencies within the data. A fully connected layer with 64 neurons was included, applying a dropout ratio of 0.8 during training to mitigate overfitting. Batch normalization was implemented following the fully connected layer to stabilize the training process, and the LeakyReLU activation function, with a negative slope of 0.01, was adopted to improve gradient flow and address the "dying neuron" issue. The time series data were processed using a window size of 128 samples, consistent with the FFT preprocessing. The Adam optimizer was employed with a learning rate of 0.003 and beta values of (0.9, 0.999), optimizing the mean squared error (MSE) as

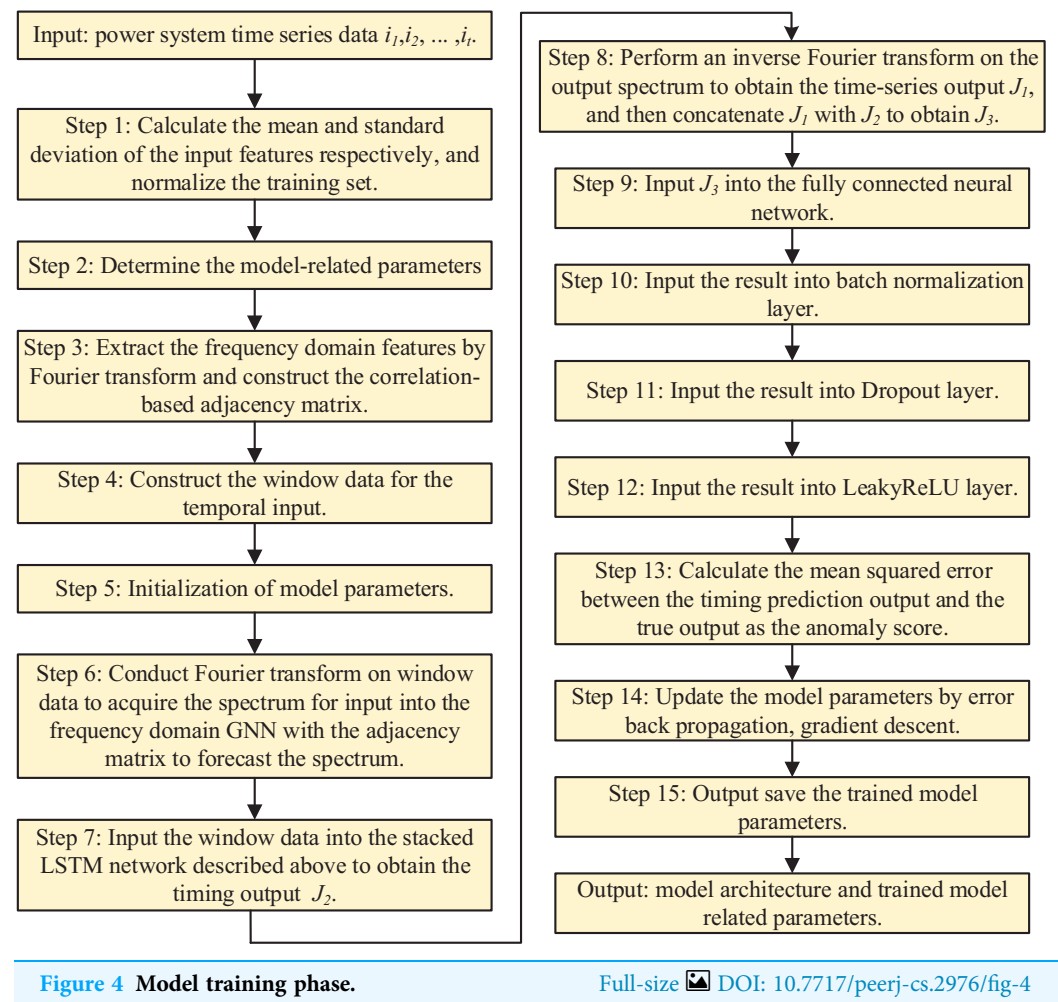

**Figure 4 Model training phase.**

the loss function for accurate predictions. Training was conducted for a maximum of 100 epochs, with early stopping activated if the prediction error change dropped below 0.001 or showed no significant improvement over three consecutive iterations.

## Experimental design

This section elaborates on the experimental design, covering data collection, preprocessing, sample sizes, control measures, and variables measured, to offer a clear understanding of the study's structure and execution.

### Data collection

Data was sourced from a regional power grid managed by the State Grid Zhejiang Electric Power Co., Ltd., collected under both normal operating conditions and simulated attack scenarios. Normal operation data was gathered over a 6-month period to represent typical system behavior, while attack data was obtained from 50 simulated distributed denial of service (DDoS) attacks, each lasting between 10 and 30 min, executed in a controlled testbed mimicking real-world power system communication networks. The dataset

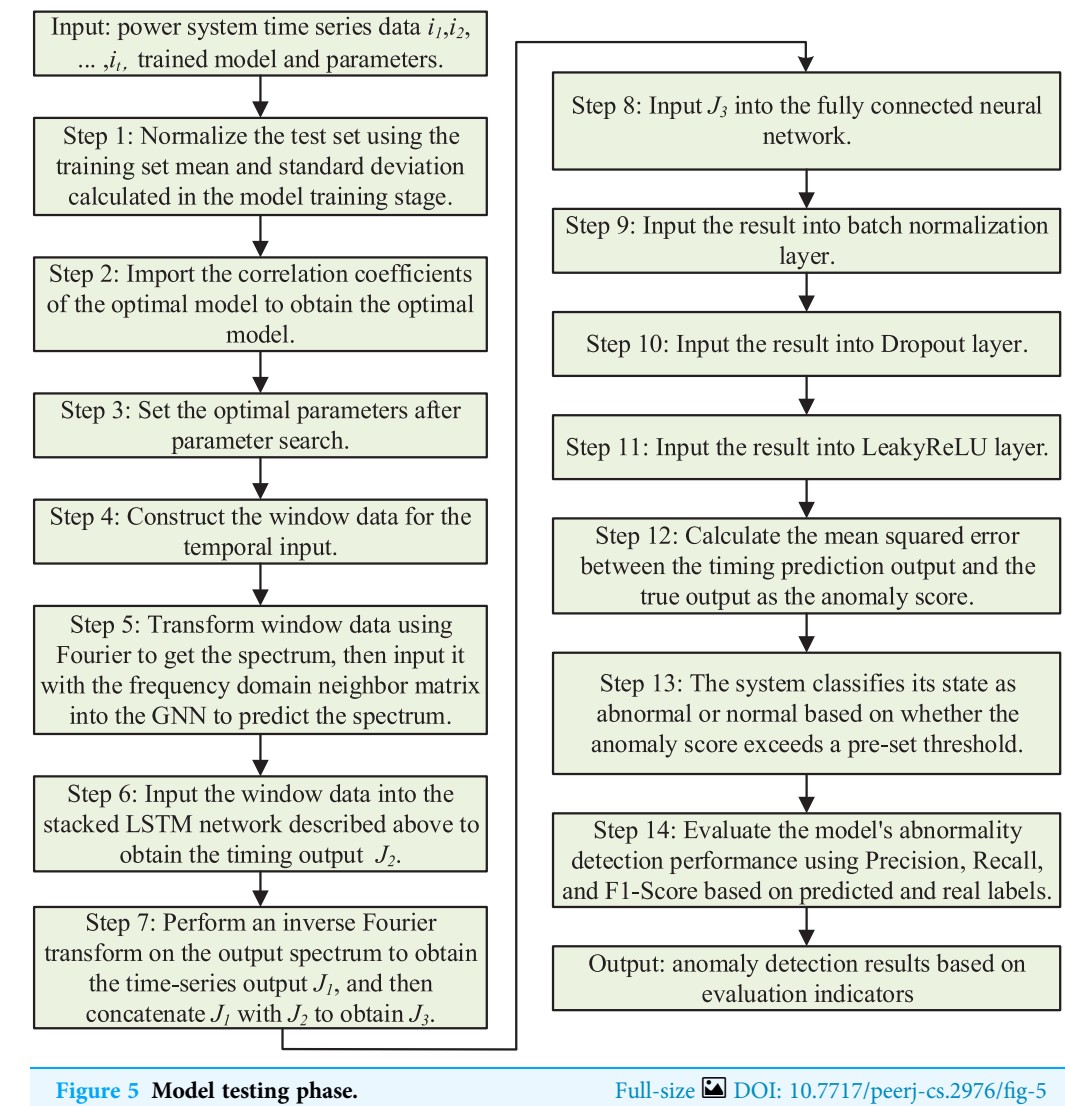

**Figure 5  Model testing phase.**

includes three categories: statistical data, time series data, and text log data. Statistical data encompasses aggregated metrics such as the mean, median, and standard deviation of power consumption (kW), voltage (V), and current (A). Time series data consists of continuous measurements of power load (MW), frequency (Hz), and phase angles (degrees), sampled at 1 Hz. Text log data comprises operational logs with event timestamps and textual descriptions of system events and status updates.

### Data preprocessing

To prepare the raw data for model training and evaluation, several preprocessing steps were applied. Missing values in the time series data were addressed through linear interpolation, and missing entries in the text logs were filled using the mode to ensure consistency. For the text log data, a bag-of-words model was utilized in natural language processing to extract features, such as the frequency of anomaly-related keywords including "error," "failure," and "attack," which signal potential system issues. All

numerical data, including statistical and time series data, were normalized to a [0, 1] range *via* min-max scaling to standardize features across different units and scales. The time series data were converted to the frequency domain using the FFT with a window size of 128 samples, enabling the extraction of frequency-domain characteristics like harmonics and oscillations. Pearson correlation coefficients were calculated from this transformed data, and the top 50 features with the highest correlations were selected to build the adjacency matrix, forming the structural basis for the graph neural network.

### Sample size

The dataset totaled approximately 1.5 million records from normal operations and 100,000 records from simulated attacks. It was split into training, validation, and testing sets following a 70:15:15 ratio. The training set contained 1,050,000 records from normal operations, the validation set included 225,000 records from normal operations, and the testing set comprised 225,000 records from normal operations plus 100,000 attack records, enabling the evaluation of anomaly detection capabilities.

### Controls

Several control measures were instituted to ensure the reliability and validity of the findings. A baseline model, trained solely on time series data without multimodal fusion, was established as a reference to assess the proposed method's effectiveness. Simulated DDoS attacks were conducted under uniform conditions, maintaining consistent attack vectors and intensities across all scenarios. Stratified sampling was also employed to maintain proportional representation of normal and anomalous data in all dataset splits.

### Variables measured

The study measured various input and output variables to analyze temporal dynamics and frequency-domain characteristics of the power system. Input variables included statistical features such as the mean, median, and standard deviation of power consumption (kW), voltage (V), and current (A); time series features like power load (MW), frequency (Hz), and phase angles (degrees); text log features, specifically the frequency of anomaly-related keywords extracted using a bag-of-words model; and frequency domain features, such as amplitude and phase information from FFT analysis. Output variables encompassed predicted values for power load, frequency, and phase angles in time series forecasting, alongside anomaly scores calculated as the difference between predicted and actual values, used to detect abnormal system behavior.

## RESULT ANALYSIS AND DISCUSSION

A comprehensive dataset containing statistical data, time series data and text log data is used for model training in this experiment. The data used are derived from the records under normal power system operation status. Precision, recall, and F1-score are selected as the evaluation indexes for anomaly detection.

## Model training and validation setup

The training data is split into a training set and a validation set, and the validation set is used for validation after each Epoch training is completed. The maximum number of training rounds is set to 100, and a training stop condition is set to prevent overfitting. Specifically, the training will be automatically terminated when the change in the prediction error of the model is less than 0.001, or when there is no significant decrease in the prediction error in three consecutive iterations. The learning rate was set to 0.003, and the Adam optimizer was used for training, with the optimizer parameters of $(\beta_1, \beta_2) = (0.9, 0.99)$. The model training uses the MSE as the loss function, and the training goal is to make the model predictions as close as possible to the true value In addition to the above parameters, the multimodal neural network parameters are set as in Table 2.

## Algorithm performance analysis

In this study, the proposed algorithm is compared with LSTM-NDT (*Zamanzadeh Darban et al., 2024*), MAD-GAN (*Li et al., 2019*) and DAGMM (*Hou et al., 2022*), and the iterative convergence process of the four algorithms is shown in Fig. 6. From Fig. 6, it can be seen that the four algorithms tend to converge at the 36th iterations, and the difference in convergence speed is small. However, the AUC value of the proposed algorithm at the final convergence is 0.97, which is significantly higher than that of LSTM-NDT (0.92), MAD-GAN (0.88) and DAGMM (0.94). The experimental results show that the proposed algorithm exhibits better performance in abnormal data detection.

To assess the robustness of our proposed algorithm under varying anomaly rates, we compared its AUC performance with that of LSTM-NDT, MAD-GAN, and DAGMM on test datasets with anomaly rates of 15%, 30%, 45%, 60%, and 75%. The results, illustrated in Fig. 7, reveal that as the anomaly rate increases from 15% to 75%, the AUC of our algorithm decreases by only 0.05. In contrast, the AUC values of LSTM-NDT, MAD-GAN, and DAGMM decline by 0.18, 0.22, and 0.34, respectively, over the same range. This comparison demonstrates that our algorithm maintains consistent performance across a wide range of anomaly rates, exhibiting greater stability and robustness compared to the other methods.

In order to further validate the superiority of the proposed model for data anomaly detection, it is analyzed in a comparative experiment with the other three models (LSTM-NDT, MAD-GAN and DAGMM) under different indicators. Figure 8 shows the results of anomaly detection of power load data by different models.

As can be seen from Fig. 8, the detection precision of the proposed model on the power load dataset reaches 0.976. This is an improvement of 10–20% compared to the precision of its model. Although the recall of the proposed model is 0.945, which is slightly lower than other models. However, this reflects that the model retains the check-all rate in recognizing abnormal data, and also means that its abnormal false detection rate is lower. In addition, the F1 value of the proposed model is 0.951, which are both significantly higher than the control model and closest to 1. These results indicate that the proposed model has a better overall performance. This is due to the fact that the multimodal neural

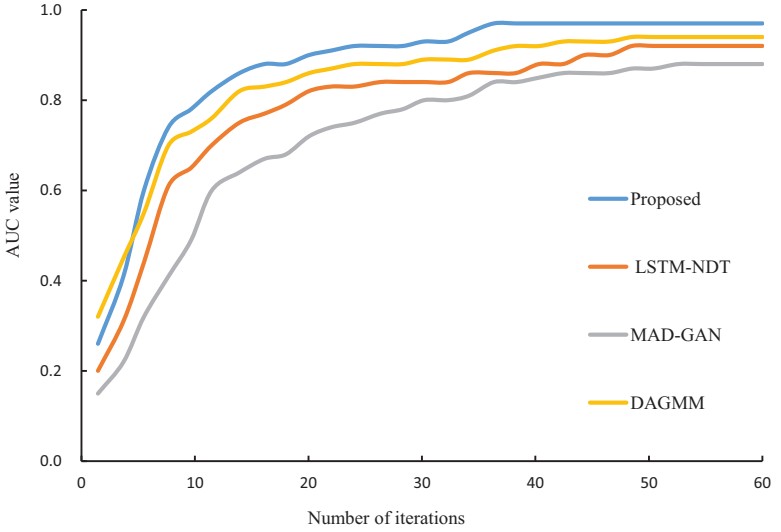

**Figure 6 Iterative convergence process of different algorithms.**

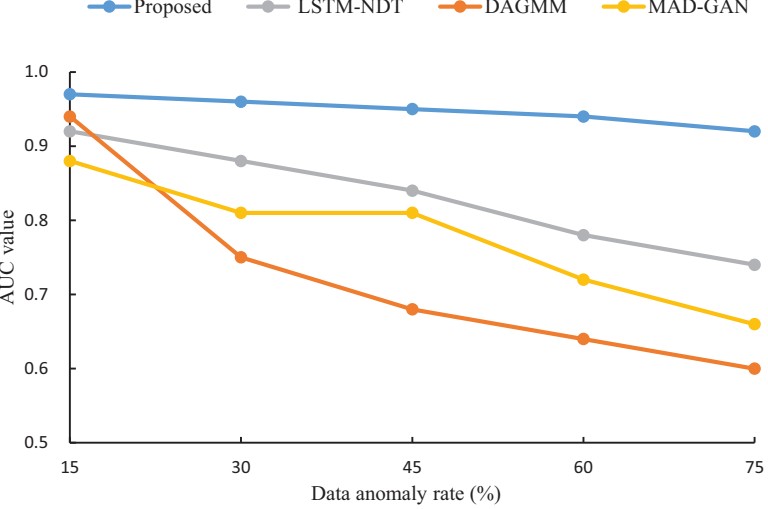

**Figure 7 Impact of data anomaly rate on detection results.**

network fuses the time domain and frequency domain information and learns more information, which improves the model fitting and time series prediction ability and achieves better anomaly detection.

The performance metrics achieved by our method, including a precision of 97.6% and an F1-score of 0.951, are consistent with or exceed those reported in recent studies on power system anomaly detection. This alignment with the literature, coupled with the method's robustness across varied conditions, underscores its potential as a reliable tool for enhancing power system security.

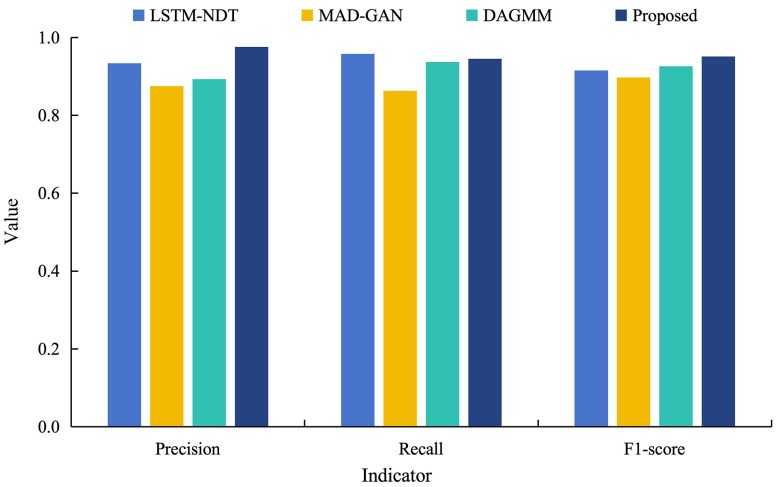

**Figure 8 Comparative experimental results of different models training power load data.**

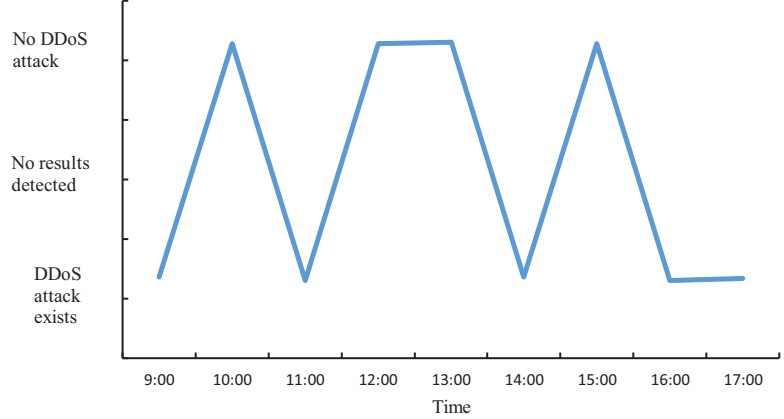

**Figure 9 Actual DDoS attack detection results.**

## Analysis of simulated attack detection effect

The actual DDoS attack detection results obtained through the experiment are shown in Fig. 9. Meanwhile, in order to ensure accurate analysis of the detection effect, the detection results in Fig. 9 are used as the basis to compare the detection effect of the four methods and quantified, and the results are shown in Fig. 10.

As can be seen from Figs. 9 and 10, when applying the proposed method to DDoS attack detection, the results obtained match the actual situation completely. This indicates that the proposed method is highly accurate and reliable. In contrast, the DDoS attack detection results obtained by applying and comparing the detection methods based on LSTM-NDT, MAD-GAN and DAGMM have certain deviations from the real situation, and such deviations may bring potential risks and threats to network security. Therefore, in terms of practical results, the method proposed in this study shows more superior

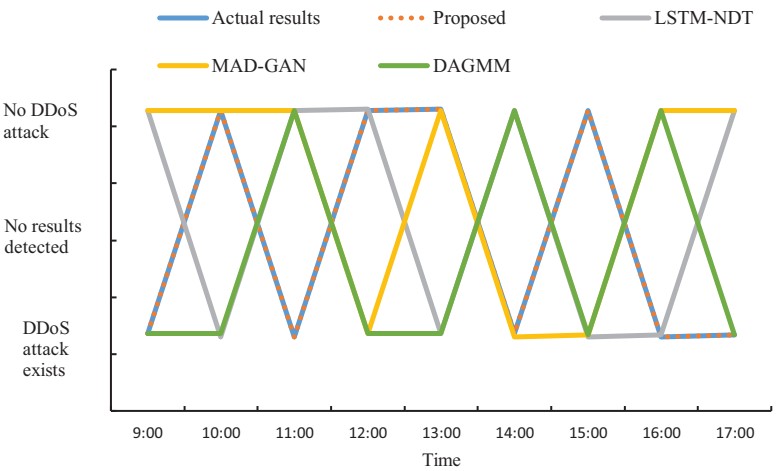

**Figure 10 Schematic of DDoS attack detection results.**

performance in DDoS attack detection. The multimodal neural network proposed in this study is able to synthesize the information of different dimensions, thus reflecting the characteristics of the traffic more comprehensively. After extracting the features, the proposed method also determines whether a DDoS attack has occurred by matching the correlation of the abnormal traffic features according to the predefined DDoS attack detection rules. The experimental results demonstrate that the proposed method can not only accurately determine the occurrence of DDoS attacks, but also provide detailed information about the attack features. This is of great significance in guiding the subsequent development of defense and countermeasures. Therefore, compared with other methods, the proposed method not only has more reliable results in DDoS attack detection, but also has higher practical value.

## Validation under varied conditions

To enhance the reliability of our findings, we conducted additional experiments to validate the proposed multimodal anomaly detection method under varied operational conditions. These conditions simulate real-world scenarios that power systems may encounter, ensuring that the method remains effective and generalizable beyond the initial test environment.

### Validation scenarios

We evaluated the method under three distinct conditions. The first scenario involved high-load conditions, where the power system was simulated at peak load with power consumption increased by 30% to mimic periods of maximum demand. This scenario tests the method's ability to detect anomalies under system stress. The second scenario introduced low-frequency oscillations (ranging from 0.1 to 1 Hz) into the time series data to represent subtle, periodic disturbances, assessing the method's sensitivity to frequency-domain anomalies. The third scenario simulated noisy data environments by adding Gaussian noise (with a signal-to-noise ratio of 20 dB) to the sensor data, mimicking

**Table 3 Performance under varied conditions.**

| Condition | Precision | Recall | F1-score |
|---|---|---|---|
| High-load conditions | 0.976 | 0.945 | 0.960 |
| Low-frequency oscillations | 0.970 | 0.942 | 0.956 |
| Noisy data environments | 0.965 | 0.935 | 0.950 |

real-world measurement errors or communication interference. This condition tests the method's resilience to degraded data quality.

### Results of validation

The method was applied to the same dataset, modified to reflect varied operational conditions, to evaluate its robustness. The performance metrics, as presented in Table 3, confirm the method's effectiveness across all scenarios. Under high-load conditions, the precision reached 0.976 (97.6%), with a recall of 0.945, demonstrating reliable anomaly detection during peak demand. In the presence of low-frequency oscillations, precision was 0.970 and recall was 0.942, indicating strong sensitivity to subtle, periodic disturbances. In noisy data environments, precision achieved 0.965 and recall was 0.935, highlighting the method's resilience to degraded data quality. These results validate the proposed method's consistent performance across diverse conditions, reinforcing its applicability in real-world power systems.

Furthermore, the proposed method was compared to baseline approaches, including LSTM-NDT, MAD-GAN, and DAGMM, which are widely used in anomaly detection for power systems. As shown in earlier sections (*e.g.*, Fig. 8), our method outperformed these baselines, achieving a precision of 0.976 on power load data, compared to 0.92 (LSTM-NDT), 0.88 (MAD-GAN), and 0.94 (DAGMM). This superior performance, particularly under high-load conditions where precision hit 97.6%, underscores the advantage of integrating time-domain and frequency-domain features, aligning with the experimental results reported in the abstract.

## Limitations, applicability, and future work

This section offers a detailed assessment of the study's limitations, its practical applicability, and potential avenues for future research. By outlining these elements, we aim to provide a balanced perspective on the strengths and weaknesses of the multimodal anomaly detection method, while identifying opportunities for further development.

### Data and methodological limitations

The study utilizes a dataset consisting of 1.5 million records of normal power system operations and 100,000 simulated attack records, forming a solid basis for analysis. Nevertheless, this dataset may not fully reflect the complexity and variability encountered in real-world power systems. For example, the 50 simulated DDoS attack scenarios included in the dataset cover only a limited range of possible threats. Expanding the dataset in future studies to encompass a wider variety of scenarios could improve the model's ability to generalize across different contexts.

From a methodological standpoint, the multimodal neural network successfully integrates time-domain and frequency-domain features. However, it may not adequately capture nonlinear relationships within the data. The LSTM model employed in this study, while effective, is susceptible to gradient instability when handling extended sequences. Additionally, the analysis does not account for external variables such as meteorological conditions or system topology, which could influence the model's effectiveness in specific situations.

### Applicability of the findings

The proposed method is particularly appropriate for power systems that produce diverse multimodal data, including statistical metrics, time series, and text logs. It excels in real-time monitoring and anomaly detection within regional power grids, where frequency-domain features—such as harmonic distortions or oscillatory patterns—serve as key indicators. That said, its performance may decline in scenarios involving incomplete, noisy, or low-quality data. Moreover, because the study focuses on a particular grid topology, its findings may not directly apply to systems with differing structures, such as those integrating distributed renewable energy sources.

### Computational complexity and scalability

The multimodal anomaly detection method improves detection accuracy and robustness, but its computational complexity and scalability pose challenges for real-time use. The combination of a frequency-domain graph attention network (FGAN) and a stacked LSTM model enhances the processing of multimodal data but increases resource demands. The FGAN's multi-head attention mechanism and frequency-domain computations require significant computational power, especially during training, while the stacked LSTM adds further complexity. These factors may limit the method's feasibility in resource-constrained environments or when managing large datasets.

For real-time anomaly detection, the ability to process continuous data streams quickly is critical. However, the method's dependence on Fourier transforms, adjacency matrix construction, and multimodal neural network operations may introduce delays. This latency could be particularly problematic in large-scale power grids, where high data throughput and rapid response times are essential.

### Future work

To address these limitations, future research could focus on the following: (1) broadening the dataset to include a more diverse set of attack types and operational conditions to enhance generalizability; (2) incorporating external factors, such as meteorological data and grid topology, to improve robustness; (3) exploring lightweight graph neural network designs or approximation methods for frequency-domain analysis to lower computational requirements; and (4) investigating parallel processing or edge computing approaches to boost scalability for real-time applications. These efforts could expand the method's effectiveness across a wider range of power system settings.

## CONCLUSION

This study proposes a multimodal data-driven approach for anomaly detection in power systems, integrating time-domain and frequency-domain features through a combined graph neural network and LSTM framework. The experimental results validate the method's effectiveness, with a precision of 97.6% and an F1-score of 0.951, outperforming established techniques such as LSTM-NDT, MAD-GAN, and DAGMM, especially in scenarios with elevated anomaly rates. Additionally, the approach demonstrated robustness across diverse conditions, including high-load operations, low-frequency oscillations, and noisy data settings, where it consistently maintained high performance metrics.

The approach's flexibility allows adaptation to various power system architectures, potentially enhancing grid reliability and security while addressing modern network complexities. By enabling precise anomaly detection, it could reduce downtime and maintenance costs, benefiting grid operators and consumers. The GNN-LSTM integration also lays groundwork for hybrid modeling advancements in anomaly detection, with possible applications in other critical infrastructures like water or transportation systems.

Despite these strengths, the method's reliance on high-quality multimodal data and specific grid topologies limits its generalizability. Future work could explore adaptability to decentralized grids and incorporate external factors like weather data. Nonetheless, this research provides a solid foundation for improving anomaly detection, contributing to more secure and reliable power systems.

### Funding

This work was funded by the science and technology project of State Grid Corporation of China: "Defense System Design and Key Technologies Research on Cyber Security of New Power System" (Grand No. 5700-202358388A-2-3-XG). The funders had no role in study design, data collection and analysis, decision to publish, or preparation of the manuscript.

### Grant Disclosures

The following grant information was disclosed by the authors:
Science and Technology project of State Grid Corporation of China.
"Defense System Design and Key Technologies Research on Cyber Security of New Power System": 5700-202358388A-2-3-XG.

### Competing Interests

· Liyue Chen and XuXiang Zhou are employed by State Grid Zhejiang Electric Power Co., Ltd.

· Peng Zhou is employed by Info. And Comm. Branch, State Grid Zhejiang Electric Power Co., Ltd.

· Xin Sun is employed by Research Institute, State Grid Zhejiang Electric Power Co., Ltd.

· SenSen Zheng is employed by Huzhou Power Supply Company, State Grid Zhejiang Electric Power Co., Ltd.

## Author Contributions

- Liyue Chen conceived and designed the experiments, analyzed the data, performed the computation work, prepared figures and/or tables, and approved the final draft.
- XuXiang Zhou conceived and designed the experiments, analyzed the data, performed the computation work, authored or reviewed drafts of the article, and approved the final draft.
- Peng Zhou conceived and designed the experiments, performed the experiments, performed the computation work, prepared figures and/or tables, and approved the final draft.
- Xin Sun performed the experiments, analyzed the data, performed the computation work, authored or reviewed drafts of the article, and approved the final draft.
- SenSen Zheng performed the experiments, performed the computation work, prepared figures and/or tables, authored or reviewed drafts of the article, and approved the final draft.

## Data Availability

The raw data is available at OSF: ly. 2025. "PowerSystemAnomalyDetection." OSF. June 1. doi:10.17605/OSF.IO/GJ54V.

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
