# Peer review of "Anomaly detection method for power system information based on multimodal data"

_PeerJ Computer Science, doi:10.7717/peerj-cs.2976_

## Round 0.1 · original submission · Major Revisions

This paper presents an LSTM-based multimodal anomaly detection method for power systems, integrating time-domain and frequency-domain data. Reviewers agree that the approach is promising; however, several key areas require revision to improve clarity, reproducibility, and impact before being considered for acceptance.

In particular:

- Clarifying the contribution: The introduction should better differentiate this study from existing work, emphasizing how multimodal data fusion overcomes prior limitations. Incorporating recent references would strengthen its positioning.
- Expanding literature review: A comparative table summarizing existing anomaly detection methods (datasets, techniques, strengths, weaknesses) would enhance clarity. Justify the selection of LSTM-NDT, MAD-GAN, and DAGMM for comparison.
- Methodology and reproducibility: Explain why specific parameters (e.g., attention heads, LSTM layers) were chosen. Adding pseudocode or algorithmic steps would improve reproducibility.
- Dataset details: Provide information on dataset size, source, and preprocessing steps. If proprietary, provide sufficient details to ensure reproducibility.
- Experimental design: Justify the choice of comparison models and discuss whether additional experiments were considered. A sensitivity analysis on hyperparameters would strengthen the findings.
- Limitations and generalizability: Discuss scalability, computational complexity, and real-time feasibility, as these impact practical adoption.
- Conclusion: Highlight broader implications for power system security and suggest future research directions.

We recommend that the authors thoroughly address the reviewers' suggestions to enhance their manuscript before resubmission.

·

Basic reporting

Some sections contain complex sentences that may obscure the main points.

The literature review lacks depth and fails to adequately position this study within the existing body of research: Expand the literature review to include more recent studies and clearly articulate how this research fills a specific knowledge gap.

While the methodology is described, it lacks sufficient detail for replication: Provide more comprehensive descriptions of the experimental setup, including data sources, preprocessing steps, and model parameters used.

Experimental design

The methodology section provides a general overview but lacks detailed descriptions of the experimental design. Include specific information about the experimental setup, sample size, controls, and any variables measured to allow replication of the study.

The data collection methods are not sufficiently detailed, raising concerns about their validity and reliability. laborate on the data collection process, including sources, types of data collected, and any preprocessing steps taken.

Validity of the findings

Include validation of the results using different datasets or replicate the study under varied conditions to strengthen the reliability of the findings.
Provide a thorough discussion of potential limitations, including sample size, methodological constraints, and external factors that could influence the results.
Discuss the contexts in which the findings can be applied and any factors that may limit their applicability to other settings or populations.
Ensure that conclusions are directly supported by the results, and clarify how the findings contribute to the existing body of knowledge.

·

Basic reporting

1. The paper is well-structured and written in professional English. The literature review is relevant and provides a strong foundation for the study.
2. Include more recent references and provide additional details about the dataset.

Experimental design

1. The research question is clearly defined, and the methodology is rigorous.
2. Add more details about parameter tuning and include pseudocode to enhance reproducibility.

Validity of the findings

1. The findings are robust and supported by wide-ranging experiments
2. Discuss potential limitations and evaluate the generalizability of the results to other datasets or applications.

Additional comments

This article presents a groundbreaking approach to anomaly detection in power systems. This has a great potential. Thanks for such a nice article.

Abstract:
The abstract clearly outlines the problem, methodology, and results.

1. Highlight the significance of integrating time-domain and frequency-domain data to make the contribution more explicit.
2. Add a brief mention of the practical implications for real-world power systems.

Introduction:
Good Introduction.

1. Provide additional emphasis on the novelty of multimodal data fusion and how it overcomes specific limitations in current methods.
2. Include more recent references (2023–2024) so readers know how your methods compare against that study.


Literature Review:
The section provides a comprehensive overview

1. Add a comparative summary table showcasing the strengths and weaknesses of existing methods, including clustering-based, density-based, and neural network-based approaches.
2. Any reason why certain methods, such as LSTM-NDT and MAD-GAN, were chosen for comparison in the experimental section.

Methodology:
The methodology is well-detailed, with clear steps

1. Clarify the rationale behind choosing specific parameters, such as the number of attention heads or LSTM layers.
2. Can you include pseudocode or algorithmic steps for replicating the multimodal neural network construction.


Dataset:
The dataset includes diverse multimodal data

1. Can you provide more information about dataset ex. source, size, and any preprocessing steps applied.
2. Is this dataset publicly available?
Results:
The results section is comprehensive.

1. The robustness of the model is highlighted, but include a discussion on how it performs under different anomaly rates beyond the provided metrics.

Discussion:
1. Highlight potential limitations, such as computational complexity or scalability for real-time applications.


Conclusion:
The conclusion summarizes key findings and their significance

1. Emphasize broader implications for power system management and anomaly detection technology.

Reviewer 3 ·

Basic reporting

All comments have been added in detail to the last section.

Experimental design

All comments have been added in detail to the last section.

Validity of the findings

All comments have been added in detail to the last section.

Additional comments

Review Report for PeerJ Computer Science
(Anomaly detection method for power system information based on multimodal data)

1. In the study, a long short-term memory network based method is proposed for anomaly detection in power system data.

2. In the introduction section, the importance of power systems, anomaly detection, multimodal data and its relationship with deep learning are mentioned. At the end of this section, more explanatory and itemized information should be given about the main differences of this study from other studies in the literature, its main contributions to the literature and its originality points.

3. In the state of the art section regarding the study, more detailed information should be given about the studies that include both traditional and deep learning based anomaly detection methods in the literature. For this, it is suggested to add a literature table consisting of columns such as the dataset used, originality, method, results, pluses, minuses. Then, by emphasizing this table, the pluses of this study in comparison with other studies in the literature can be expressed more clearly.

4. When the deep learning based algorithm proposed in the study, which can perform anomaly detection operations, is examined in detail, it is observed that it has a certain level of originality. Comparison of the proposed method with three other different models in the literature has increased the quality of the study. However, although there are many different deep learning-based models that can be used to solve this problem in the literature, it should be explained more clearly why comparisons were made with these models and/or whether different experiments were made.

5. Regarding the shared parameters of the multimodal neural network model given in Table-1, it is seen that the most important metrics are specified. However, since the parameter choices are very important for the model, it should be emphasized more clearly how these parameters are determined.

6. When we consider the results of the study and their comparison with the literature, it is observed that the results are at an acceptable and appropriate level.

As a result, this study proposes a method that has a high potential to contribute to the literature for anomaly detection. However, great attention should be paid to the sections listed above.

Cite this review as

---

## Round 0.2 · accepted · Accept

Reviewers unanimously agreed that all comments from the previous round of reviews have been adequately addressed, resulting in a substantial improvement in quality, and recommend the current version of the manuscript for acceptance.

·

Basic reporting

Thanks, now the main points are presented more clearly and concisely. And the authors answered all comments.

Experimental design

They have revised the manuscript to include comprehensive details about the experimental setup, sample size, controls, and variables measured, as mentioned in (“3.5 Experimental Design”).

Validity of the findings

They have added a new subsection, "4.4 Validation Under Varied Conditions," to the "Result Analysis and Discussion" section, as requested.

·

Basic reporting

This version looks good, thanks for addressing the comments. I accept this article.

Experimental design

This version looks good, thanks for addressing the comments. I accept this article.

Validity of the findings

This version looks good, thanks for addressing the comments. I accept this article.

Additional comments

This version looks good, thanks for addressing the comments. I accept this article.

Reviewer 3 ·

Basic reporting

All comments have been added in detail to the last section.

Experimental design

All comments have been added in detail to the last section.

Validity of the findings

All comments have been added in detail to the last section.

Additional comments

Review Report for PeerJ Computer Science
(Anomaly detection method for power system information based on multimodal data)

The responses to the referee comments and the changes made to the paper accordingly are generally at an appropriate level. Therefore, I recommend that the paper be accepted. I wish the authors success in their future papers. Best regards.

Cite this review as